# BioVERSE: Representation Alignment of Biomedical Modalities to LLMs for Multi-Modal Reasoning

## Abstract

Recent advances in large language models (LLMs) and biomedical foundation models (BioFMs) have achieved strong results in biological text reasoning, molecular modeling, and single-cell analysis, yet they remain siloed in disjoint embedding spaces, limiting cross-modal reasoning. We present BioVERSE (**Bio**medical **V**ector **E**mbedding **R**ealignment for **S**emantic **E**ngagement), a two-stage approach that adapts pretrained BioFMs as modality encoders and aligns them with LLMs through lightweight, modality-specific projection layers. The approach first aligns each modality to a shared LLM space through independently trained projections, allowing them to interoperate naturally, and then applies standard instruction tuning with multi-modal data to bring them together for downstream reasoning. By unifying raw biomedical data with knowledge embedded in LLMs, the approach enables zero-shot annotation, cross-modal question answering, and interactive, explainable dialogue. Across tasks spanning cell-type annotation, molecular description, and protein function reasoning, compact BioVERSE configurations surpass larger LLM baselines while enabling richer, generative outputs than existing BioFMs, establishing a foundation for principled multi-modal biomedical reasoning.

## 1 Introduction

High-throughput assays such as scRNA-seq, proteomics, and small-molecules profiling generate rich, high-dimensional data that are critical for biomedical discovery. Biomedical foundation models (BioFMs; also referred to as BMFMs) trained on those inputs, e.g., scGPT (Cui et al., 2024) for single-cell RNA sequencing (scRNA-seq), ESM-2 (Lin et al., 2023) for proteins, Molformer (Ross et al., 2022) for small molecules, capture expressive representations but lack instruction-following and open-ended reasoning. In contrast, general-purpose large language models (LLMs) excel at language interaction and can nominally ingest sequences like proteins or Simplified Molecular Input Line Entry System (SMILES) strings, but tokenization yields short, uninformative fragments and they cannot parse modalities such as scRNA-seq, where a cell's gene expression vector cannot be represented meaningfully as a token sequence. Bridging these strengths requires a framework that preserves modality-specific encoders, aligns their embeddings with the LLM token space, and enables reasoning across them.

We introduce BioVERSE, a framework adapting the familiar vision–language paradigm (e.g., Flamingo (Alayrac et al., 2022), BLIP-2 (Li et al., 2023), LLaVA (Liu et al., 2023a)), and more recently, InternVL3.5 (Wang et al., 2025b), to the biomedical domain. BioVERSE follows a BioFM-adapter-LLM design: it projects BioFM embeddings into the LLM's embedding space via a lightweight MLP adapter and injects them as special tokens (e.g. `[BIO_1]`, `[BIO_2]`, ..., `[BIO_k]`, and `[TRAINABLE_BIO]`). By placing biological and textual information in a shared space, BioVERSE enables joint multi-modal reasoning while directly exploiting the LLM's native memory and inference abilities. Our contributions are:

- **Modular architecture:** Plug-and-play biological encoders (scRNA-seq, protein, molecule) connect to a decoder-only LLM via a small projection layer and LoRA adapters.
- **Alignment via contrastive learning:** We directly align encoder embeddings to the language token space, i.e., no separate bio-language encoder, enabling zero-shot transfer across modalities.

- **Multimodal instruction tuning:** We curate paired (embedding, instruction, response) data so the LLM learns to use biological context in generation.

- **Practicality:** Compact BIOVERSE variants match or exceed larger baselines on joint bio–text tasks and support privacy-preserving, on-prem deployments. This aligns with the current trend of promoting small language models in the agentic AI systems (Belcak et al., 2025).

## 2 RELATED WORK

**Biomedical encoders.**   Large-scale BioFMs have been developed for modality-specific data. For transcriptomics, models such as scBERT (Yang et al., 2022), Geneformer (Theodoris et al., 2023), and scGPT (Cui et al., 2024) capture cellular states and gene–gene dependencies, with BMFM-RNA (Dandala et al., 2025) providing a reproducible framework for pretraining. For proteins, ProteinBERT (Brandes et al., 2022) and ESM-2 (Lin et al., 2023) learn contextual embeddings that support function and family prediction, while AlphaFold (Jumper et al., 2021) and ESMFold (Lin et al., 2023) show how such embeddings enable structure prediction. In the molecular domain, ChemBERTa (Chithrananda et al., 2020) and MolFormer (Ross et al., 2022) encode SMILES strings and molecular graphs into embeddings of chemical properties. Together, these unimodal encoders yield strong representations but lack natural-language reasoning.

**Bio-LLM integration.**   Several efforts have explored bridging biological embeddings with language models; however, current methods only partially address the challenge of joint bio-text reasoning. GenePT (Chen & Zou, 2024) pools gene-level embeddings derived from ChatGPT descriptions into cell-level representations, which work well for classification but are not integrated into an LLM 's generation pipeline. CELLama (Choi et al., 2024) prompts LLMs with transcriptomic profiles converted into text-like inputs. While effective for flexible queries, it does not exploit pretrained BioFMs trained directly on raw scientific data, limiting its ability to capture domain-specific signal. scCello (Yuan et al., 2024) and scMulan (Bian et al., 2024) incorporate text labels or metadata as supervision to improve biological embeddings, but the resulting embeddings remain modality-specific and are not aligned with text embeddings from LLMs, limiting their use for joint bio–text reasoning. CellWhisperer (Schaefer et al., 2024) uses Geneformer (Theodoris et al., 2023) and BioBERT (Lee et al., 2020) to align scRNA-seq and text embeddings. While this enables retrieval, differences in tokenization and architecture prevent seamless integration with generative LLMs, leading to a RAG-style pipeline rather than embedding-aware reasoning. TxGemma (Wang et al., 2025a), BioT5 (Pei et al., 2023), and Galactica (Taylor et al., 2022) fine-tune general-purpose or biomedical LLMs on biological sequences tokenized as text (e.g., amino acids, SMILES, or curated biomedical corpora). This design enables strong domain-specific reasoning and therapeutic applications but constrains the models to operate entirely in the text-token space. As a result, they do not leverage pretrained BioFMs trained on raw molecular or cellular data, limiting their ability to capture low-level biological signals and reducing extensibility across modalities. MAMMAL (Shoshan et al., 2024) unifies multiple bio-modalities and supports generation in a T5-style foundation model trained end-to-end on diverse data, but its monolithic nature and custom tokenizer preclude modular embedding reuse or deployment within modern instruction-tuned LLMs.

**General multi-modal LLMs.**   In the general AI domain, vision-language models demonstrate how non-text modalities can be modularly aligned with LLMs. Approaches like LLaVA, BLIP-2, Flamingo, and more recently InternVL 3.5 (Liu et al., 2023a; Li et al., 2023; Alayrac et al., 2022; Wang et al., 2025b) established a design pattern where modality-specific encoders are efficiently connected to LLMs through projection and instruction tuning. This pattern has yet to be fully realized in biomedicine, where biological and text embeddings remain misaligned.

**Positioning of BIOVERSE.**   Building on the proven encoder-projector-LLM design pattern from vision-language models, BIOVERSE addresses the gap between biological and textual embedding spaces by projecting BioFM outputs directly into the LLM's input embedding space. This modular alignment enables pretrained encoders for scRNA-seq, proteins, or molecules to be integrated without retraining the LLM. By treating these embeddings as first-class tokens, BIOVERSE allows the model to reason jointly over biological data and natural language, providing a flexible foundation for cross-modal biomedical intelligence.

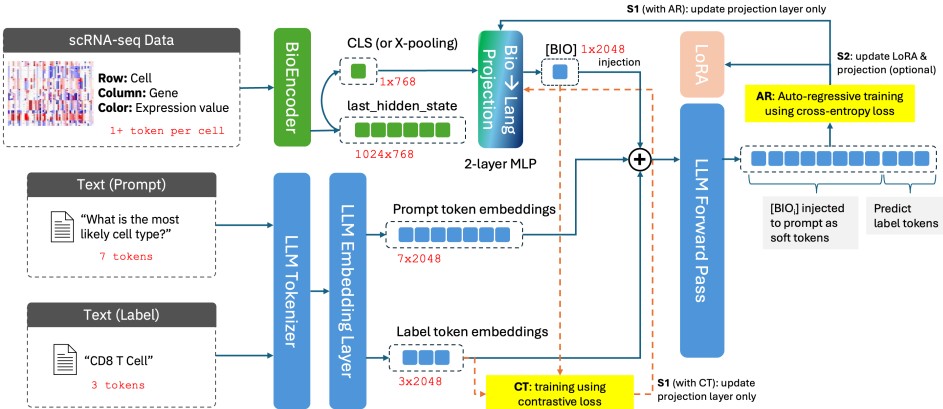

Figure 1: BIOVERSE base architecture: a modality-specific BioFM encodes a biological entity, and its output embeddings are mapped by a projection layer into the LLM's embedding space via special tokens (e.g. [BIO]). In the alignment stage, only the projection layer $P_\theta$ is trainable, while the encoder $f_b$ and the LLM $g$ remain frozen. In the subsequent instruction-tuning stage, we allow both $P_\theta$ and the low-rank adapter (LORA) within the LLM to be trainable. Stage 1 (S1) can be trained using autoregressive (AR) or contrastive (CT) loss, while stage 2 (S2) is always AR.

## 3 METHOD

### 3.1 PROBLEM SETUP

Given a biological input $x_b$ (e.g., a protein sequence or an scRNA-seq profile) and a natural-language context or query $q$, our goal is to enable a frozen LLM to jointly reason over $(x_b, q)$. We use a pretrained BioFM $f_b$ to encode $x_b$ into one or more embeddings $z_b$, and a lightweight projection $P_\theta$ to map $z_b$ to the LLM's token-embedding space. These projected embeddings are injected at designated marker positions, e.g., [BIO], and act as soft tokens that the frozen decoder can attend to. The key challenge is that bio embeddings and text embeddings are trained in siloed spaces and must be aligned for effective joint reasoning.

### 3.2 TWO-STAGE TRAINING

**Alignment** Although LLM can, in principle, learn cross-space attention through sufficient instruction tuning, pre-aligning the bio and language embeddings provides a strong inductive prior: it reduces task-specific tuning burden and improves zero/few-shot generalization to unseen tasks. To achieve this, we introduce a CLIP-style alignment stage using paired data $(x_b, t_b)$, where the projection $P_\theta$ is trained so that the bio embedding $z_b = f_b(x_b)$ is close to its language counterpart $\phi(t_b)$, which represents the text's embedding.

In the base training mode illustrated in Figure 1, all data is processed through LLM's forward pass, and a standard autoregressive cross-entropy loss (with teacher forcing) is used to guide learning. Alternatively, to avoid the costly forward pass required by a large LLM, and to enable alignment with efficient encoder models that are often co-trained with the decoder for retrieval, we evaluate an alternative alignment mode. In this variant, we use contrastive learning to directly align the bio embeddings with their paired text embeddings. From here onward, we denote the first alignment strategy as AR (autoregressive) and the second as CT (contrastive).

**Instruction Tuning** The alignment stage maps BioFM embeddings into the LLM's token space; the instruction tuning stage teaches the decoder to use those soft tokens under real prompts, improving generative reasoning, prompt robustness, and likelihood calibration. Abundant curated corpora (e.g., TxGemma-processed instruction sets and Therapeutics Data Commons (TDC) tasks) can be readily adapted for multi-task SFT, making this step practical with minimal data engineering. However, instruction tuning can confound alignment comparisons, increase compute and hyperparameter burden, and cause geometry drift. Given that our focus is alignment, we do not perform extensive instruction tuning.

### 3.3 Modular Architecture

**Biological Encoder**    We use a pretrained BioFM $f_b : \mathcal{X}_b \to \mathbb{R}^{k \times d_b}$ to encode a biological input $x_b$ into $k$ embeddings:[1]

$$z_b = f_b(x_b), \quad z_b \in \mathbb{R}^{k \times d_b}$$

Any BioFM appropriate to the modality can be used, provided a bio embedding can be extracted from its output or an intermediate layer. We refer to it as an encoder to highlight its role in mapping a biological entity into an embedding, although the underlying architecture may be an encoder or a decoder. Some BioFMs return a single pooled embedding (e.g., scGPT (Cui et al., 2024), scBERT (Yang et al., 2022)), while others output a sequence of contextual embeddings (e.g., per-residue embeddings in ESM-2 (Lin et al., 2023), or per-token embeddings over SMILES in ChemBERTa (Chithrananda et al., 2020). In practice, these sequence outputs are often pooled to obtain a single vector per entity, but our framework supports both pooled and multi-token cases.

**Projection Layer**    The projection $P_\theta : \mathbb{R}^{d_b} \to \mathbb{R}^{d_t}$ is a lightweight MLP with ReLU activations, layer normalization, and dropout for stability. It maps the BioFM output into the LLM embedding space:

$$\tilde{z}_b = P_\theta(z_b), \quad \tilde{z}_b \in \mathbb{R}^{k \times d_t}$$

In vision-language models, tens to hundreds of tokens are used per image (e.g., BLIP-2 (Li et al., 2023), CLIP (Radford et al., 2021)), often requiring token-level normalization or gating for stability. By contrast, BioFMs usually pool to a single token, reflecting that cells, proteins, and molecules are typically treated as indivisible units, and a lightweight projection is sufficient to ensure compatibility with the LLM while preserving BioFM semantics.

**Injection of Bio Tokens**    We inject the projected bio embeddings $\tilde{z}_b$ at a placeholder (e.g., `[BIO]`) within the query $q$, replacing the marker with the embeddings as soft tokens before concatenating with the rest of the sequence:

$$[\,\text{Tokens}(q, \texttt{[BIO]} \to \tilde{z}_b)\,;\ \text{Tokens}(t_b)\,].$$

Here, Tokens$(\cdot)$ denotes text after tokenization and embedding lookup. While standard text inputs are mapped from token IDs through the embedding matrix, projected bio embeddings are fed directly into the LLM embedding layer (via the `inputs_embeds` interface in many implementations), enabling integration without modifying the tokenizer or embedding matrix.

**Language Embedding Targets**    When performing alignment training using AR loss, the model consumes Tokens$(t_b)$ directly, since the objective is next-token prediction over a text sequence. However, to also support CT loss that enforces representation-level similarity, we require a single pooled representation of the text. We therefore define a frozen language embedding $\phi(t_b) \in \mathbb{R}^{d_t}$ extracted from the target LLM. Choices of the target will be discussed later in detail.

**Language Model**    We extend a small LLM $g$ with a few soft tokens, without modifying its tokenizer or positional encodings. As the lightweight projection layer decoupled the LLM and requires only embedding dimension compatibility, our approach can directly scalable to larger LLMs.

### 3.4 Alignment Objectives

**Autoregressive Decodability.**    Our default alignment strategy is to directly train the LLM to use the projected bio embeddings during generation. Given a query $q$, bio embeddings $\tilde{z}_b$ injected at a `[BIO]` marker, and paired target text $t_b = (t_1, \ldots, t_{|t_b|})$, we minimize the negative log-likelihood of predicting $t$ in an autoregressive manner:

$$\mathcal{L}_{\text{AR}} = -\sum_{i=1}^{|t_b|} \log p_{\text{LLM}}\big(t_i \mid \tilde{z}_b, q, t_{<i}\big)$$

This objective explicitly teaches the LLM to attend to bio tokens in the same way it attends to text tokens, ensuring decodability and downstream reasoning ability. Because the loss is defined over natural text generation, it tightly couples alignment with the LLM's causal decoding process.

---

[1]For notational simplicity, we assume $k = 1$ and omit the token index in most equations from here onward; the framework naturally extends to $k > 1$ when multiple embeddings are injected

**Contrastive Alignment.** In addition to autoregressive decodability, we also study a contrastive alignment mode that enforces representation-level similarity. Here, the projected bio embeddings $\tilde{z}_b$ are aligned with text embeddings $\phi(t_b)$ from paired descriptions using a bidirectional InfoNCE loss:

$$\mathcal{L}_{\text{CT}} = -\frac{1}{2N} \sum_{i=1}^{N} \left[ \underbrace{\log \frac{\exp(\text{sim}(\tilde{z}_b^{(i)}, \phi(t_b^{(i)}))/\tau)}{\sum_{j=1}^{N} \exp(\text{sim}(\tilde{z}_b^{(i)}, \phi(t_b^{(j)}))/\tau)}}_{\text{bio} \to \text{text}} + \underbrace{\log \frac{\exp(\text{sim}(\phi(t_b^{(i)}), \tilde{z}_b^{(i)})/\tau)}{\sum_{j=1}^{N} \exp(\text{sim}(\phi(t_b^{(i)}), \tilde{z}_b^{(j)})/\tau)}}_{\text{text} \to \text{bio}} \right]$$

where $\text{sim}(\cdot, \cdot)$ denotes cosine similarity and $\tau$ is a learnable temperature. Note that the denominator from bio to text normalizes over all text embeddings, and the denominator from text to bio normalizes over all bio embeddings. Prior work (e.g. CLIP (Radford et al., 2021), BLIP-2 (Li et al., 2023)) shows including both directions stabilizes training.

We explore contrastive alignment for three main reasons: (1) it enforces semantic consistency between bio and text embeddings rather than relying solely on next-token prediction, which may improve generalization to unseen tasks; (2) it decouples alignment from the frozen LLM decoder, allowing alternative text encoders to serve as alignment targets; and (3) it is computationally efficient, bypassing the LLM's forward pass to directly align paired $(x_b, t_b)$ examples. An additional benefit, observed in prior work, is that contrastive objectives produce more isotropic embedding spaces and can exploit large in-batch negatives, improving transfer and data efficiency.

## 4 EXPERIMENTAL SETUP

### 4.1 MODELS

**Biological Encoder** We evaluate representative foundation models across three modalities (all pooled into a single embedding at the end): scGPT (Cui et al., 2024) for scRNA-seq, ESM-2 (Lin et al., 2023) for proteins, and ChemBERTa (Chithrananda et al., 2020) for small molecules. We also include MAMMAL (Shoshan et al., 2024), a multimodal biomedical model that supports all three modalities. Finally, we consider a general-domain LLM used directly as a bio encoder. Although LLMs can ingest serialized versions of biological entities (e.g., scRNA-seq approximated by sorting genes into a sequence, while proteins and SMILES strings are natively sequential), the resulting tokenizations tend to be short and poorly contextualized, and we suspect it limits biological fidelity.

**Language Model** We evaluate two scales of LLMs as the language backbone. As a small model, we use Granite-3.3-8B-Instruct (Granite-8B for short), an 8B open-weights model released by IBM Research, to demonstrate alignment effectiveness under limited capacity. BIOVERSE is LLM-agnostic: any model that accepts embedding inputs (as is the case for most HuggingFace LLMs) can be used without architectural changes.[2] For comparison against large-scale baselines that do not leverage BioFMs for encoding, we also evaluate GPT-OSS-120B, a public 120B open-weights model by OpenAI, details of models see App. A.2)..

**Projection Layer** The projection is implemented as a three-layer MLP with ReLU activations, layer normalization, and dropout for stability.

**Language Embedding Target** We evaluate four ways to construct $\phi(t)$: (1) **TokEmbed**, averaging input embeddings; (2) **LL-Mean**, mean pooling of the final layer; (3) **LayerAvg**, averaging several top layers; and (4) **LLM-Embed**, a co-trained text encoder when available. TokEmbed performed poorly, and we adopt LL-Mean as the default. LLM-Embed shows strong results but applies only under contrastive training; systematic study of LayerAvg and LLM-Embed is left for future work.

### 4.2 DATASETS AND EVALUATION

#### 4.2.1 ALIGNMENT

For alignment, we construct paired biological entities $(x_b)$ and textual descriptions $(t_b)$ across three modalities. **Protein:** We obtain protein–text pairs from UniProtKB, where each amino acid sequence

---

[2]In practice, this requires only that the LLM expose an `inputs_embeds` interface or equivalent.

is linked to curated Gene Ontology (GO) terms representing its functional annotations across the three GO namespaces: Biological Process, Molecular Function, and Cellular Component. GO term metadata is derived from the official GO ontology, and annotations are obtained from UniProt cross-references. To ensure reliability, we retain only experimentally supported GO annotations, yielding high-quality supervision for aligning BioFM protein embeddings with language representations. **Small Molecule:** For small molecules, we leverage LLASmol (Yu et al., 2024), which provides SMILES–text pairs with chemically grounded descriptions. Specifically, we select two datasets from the LLASmol collection for BioFM alignment: SMILES-to-IUPAC conversion and molecule captioning. In both cases, each molecule is represented as a SMILES string paired with natural-language annotations of structure, properties, or activities. **scRNA-seq:** For single-cell data, we adopt CellWhisperer (Schaefer et al., 2024), which aligns scRNA-seq profiles with cell-type and tissue-level textual metadata. Following the dataset protocol, we use the CellxGene subset (Perkel, 2024), where pseudo-bulk RNA samples are generated by averaging single-cell profiles, and natural-language descriptions are produced from cell and tissue metadata using large language models. This enables alignment between transcriptomic embeddings and ontological descriptions.

### 4.2.2 INSTRUCTION TUNING

For Stage 2 instruction tuning, we augment the alignment dataset with templated prompts paired to the biological–text examples. This teaches the LLM to use aligned bio tokens under user queries; for example, *"What cell type matches this* `[BIO]` *gene-expression profile?"* Since the primary goal of this paper is to evaluate architectural design rather than extensive prompt handling, we limit instruction tuning to light augmentation. This procedure can be readily extended with training data from prior works such as TxGemma (Wang et al., 2025a), which introduces instruction-style data for therapeutic reasoning tasks using the Therapeutics Data Commons (TDC), and CellWhisperer (Schaefer et al., 2024) for cell-related tasks.

### 4.3 EVALUATION

We evaluate our approach on six downstream tasks: five from Mol-Instruct (Fang et al., 2023) (four protein-related and one small-molecule) and one from scEval (Liu et al., 2023b) (cell-type annotation). Mol-Instruct provides molecular question–answer pairs spanning property prediction, reaction reasoning, and therapeutic relevance, while scEval offers benchmarks for scRNA-seq applications. For generative tasks, we report results using three complementary metrics. **LLM-as-a-judge:** GPT-OSS-120B scores each model response independently against the expected output with single-output, reference-based prompt (see Appendix A.4, repeating each evaluation three times under different random seeds. **BERTScore:** captures semantic similarity. **ROUGE-L:** measures surface-form overlap. Full definitions of the metrics and the prompt used for LLM-as-a-judge are provided in Appendix A.1. Training details (learning rate, batch size, optimizer, training duration, and compute resources) are documented in Appendix A.3.

## 5 RESULTS

### 5.1 EMBEDDING ALIGNMENT VISUALIZATION

To illustrate the effect of our alignment procedure, we present UMAP projections of scRNA-seq embeddings and their corresponding natural language embeddings before and after applying BioVERSE alignment. As shown in Figure 2, prior to alignment, the two modalities occupy largely disjoint regions of the latent space, whereas after training the projection layer they exhibit clear overlap, indicating successful cross-modal alignment. These visualizations serve as a qualitative preview of BioVERSE's capacity to unify biological and textual representations.

### 5.2 MAIN RESULTS

### 5.2.1 ZERO-SHOT GENERATIVE CELL TYPE ANNOTATION

We evaluate BIOVERSE's against two baselines on the PBMC10K dataset as discussed in scvi-tools (Gayoso et al., 2022) under zero-shot generation: (1) random and majority baseline (2) open-domain

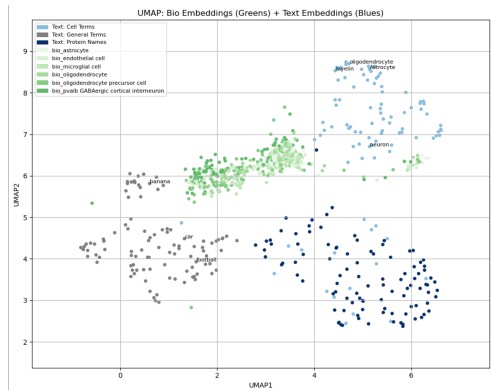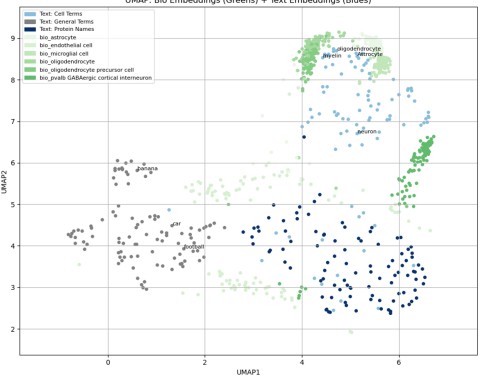

Figure 2: UMAP visualization of scRNA-seq and text embeddings. Left: before alignment, cell embeddings (green) form isolated clusters within the LLM embedding space. Right: after alignment, cell embeddings are pulled closer to biologically relevant text and separated from unrelated general-domain text. BIOVERSE successfully realigns the modalities into a shared representation space.

Table 1: Zero-shot PBMC10K results with 9 cell types.

|  | Baseline | | Matching | Generative | | |
|  | Random | Majority | LangCell | Granite-8B | GPT-OSS-120B | BIOVERSE |
|---|---|---|---|---|---|---|
| Accuracy | 0.111 | 0.417 | 0.865 | 0.369 | 0.779 | 0.614 |
| Macro $F_1$ | 0.086 | 0.065 | 0.896 | 0.262 | 0.543 | 0.437 |

LLMs given a list of the 128 most expressed genes (sorted by expression count) as input. Alignment is trained on CellxGene data aggregated into pseudo-bulk samples as in CellWhisperer (Schaefer et al., 2024). The PBMC10K dataset used for evaluation is not present in CellxGene; however, CellxGene contains another PBMC dataset among its 1,800+ scRNA-seq datasets. Thus, while the exact test set is excluded, the ontology of cell types is shared. This reflects a realistic zero-shot transfer setting.

As shown in Table 1, although majority voting achieves relatively high accuracy, it fails on minority classes, leading to poor macro-$F_1$. Prior BioFMs such as scGPT (Cui et al., 2024) and alignment-based models like LangCell (Zhao et al., 2024) and scMMGPT (Shi et al., 2025), when performing cell type annotation under zero-shot setting, fundamentally operate in a candidate-space matching paradigm. These models project cells and a predefined set of candidate labels and their descriptions into a shared embedding space and assign the nearest match. LangCell achieves the highest scores, reflecting the relative ease of candidate-space matching. By contrast, generative models operates in a generative regime: the LLM must produce a natural language label rather than selecting the nearest candidate. In our setup, we apply prompt-level constraints, instructing the model to select only from a predefined option set without decoding-level enforcement. The model nevertheless engages in open-ended reasoning before aligning to a candidate, making the task inherently more difficult. This setting, however, offers unique advantages: the ability to articulate rationales, propose novel labels outside a fixed ontology, and integrate bio-embeddings with broader biomedical knowledge.

Open-domain LLMs perform substantially better than chance, indicating that even with only sorted gene lists, LLMs show some inherent capability for this task. BIOVERSE improves substantially over its backbone (Granite-8B) while preserving and enhancing the LLM's reasoning ability when both `[BIO]` and gene-list evidence are provided in the prompt. The `[BIO]` token guides the model toward the correct type, but crucially also anchors the explanation to biological features, yielding more faithful rationales than using gene lists alone. While overall accuracy still trails candidate-matching approaches, the generative setting enables richer outputs: models can articulate why a label was chosen, highlight relevant genes, and remain extensible to novel types outside a fixed ontology. Future work will explore strengthening this interpretive capacity (e.g., through multiple `[BIO]` tokens tied to specific pathways or gene modules) and scaling aligned projections to larger LLMs.

```
True Label: CD14+ Monocytes
Predicted Label: Based on the sorted expressed genes, the most likely immune cell subtype
is CD14+ Monocytes. The presence of genes such as TYROBP (DAP12), FCER1G (FcgRI), ITGB2
(CD29), and ITGAM (CD11b) suggests a monocytic lineage. [...skip] The absence of B cell-
specific genes and T cell receptor genes (TR genes) further supports this conclusion.
```

Figure 3: Example generative annotation on PBMC10K: BIOVERSE produces the label and reasoning grounded in gene evidence.

Table 2: Molecular description generation results. S1: projection-only. S2: projection+LoRA

| Model | BioFM | S1 | S2 | LLM-J | BERT-S | ROUGE-L |
|---|---|---|---|---|---|---|
| BIOVERSE | MAMMAL | 30k | 30k | **0.17** | **0.92** | **0.20** |
| | | 30k | – | 0.10 | 0.92 | 0.18 |
| BIOVERSE | ChemBERTa | 130k | – | 0.10 | 0.91 | 0.18 |
| | | 30k | – | 0.08 | 0.90 | 0.16 |
| Granite-8B | | | | 0.04 | 0.91 | 0.07 |
| LLaMA-70B | (not applicable) | | | 0.05 | 0.90 | 0.06 |
| Mixtral-8x7B | | | | 0.05 | 0.91 | 0.08 |
| GPT-OSS-120B | | | | 0.02 | 0.89 | 0.06 |

### 5.2.2 MOLECULE DESCRIPTION GENERATION

The molecular description generation task in Mol-Instructions (Fang et al., 2023) evaluates a model's ability to produce detailed free-text descriptions of molecules given their SMILES representation. Target outputs cover structural features, physicochemical properties, biological activities, and potential applications, requiring the model to bridge symbolic chemical notation with natural language. We compare BIOVERSE with open-weight LLMs ranging from 8B (the same size as the BIOVERSE backbone) to 120B, all without BioFM alignment and therefore relying only on raw tokenized SMILES strings. We also test on the effect of using different BioFMs (ChemBERTa vs. MAMMAL) to generate the initial molecular embeddings. This tests whether BIOVERSE can flexibly adapt to modality-specific encoders without losing stability. All evaluations are conducted in a *zero-shot transfer* setting: Mol-Instructions descriptions are not used during alignment. Instead, BIOVERSE is aligned on independent molecule–text pairs from LLASMol (Yu et al., 2024), as described in Section 4.2.1. Table 2 shows BIOVERSE outperform open-domain LLMs significantly, regardless of the size. Switching from MAMMAL to ChemBERTa yields slightly worse results under the same training iterations, indicating that the framework is plug-and-play and stable across different molecular encoders. Additionally, the two-stage strategy (S1 followed by S2) is more effective than simply training S1 for longer. All three evaluation metrics show a consistent trend across our tests. We consider LLM-J to be the most meaningful metric for free-text generation; we therefore report only this metric in subsequent results.

### 5.2.3 PROTEIN-ORIENTED TEXT GENERATION

We evaluate all four protein-oriented text generation benchmarks from Mol-Instructions (Fang et al., 2023): (1) catalytic activity prediction, (2) domain/motif prediction, (3) functional description generation, and (4) protein function prediction. Each task provides a protein sequence as input, and the model must generate free-text outputs describing a specific property of that sequence. Together, these tasks probe both factual grounding (e.g., motif recognition) and open-ended description ability, testing whether the model can jointly reason over the protein sequence and the accompanying prompt.

Similar to the molecular task, we compare BIOVERSE with open-weight LLMs without BioFM alignment and therefore relying only on raw tokenized amino-acid sequences. We also conduct self-comparisons along two axes: (1) training iterations and (2) alignment strategies. As shown in Table 3, across all four tasks, BIOVERSE consistently outperforms open-domain LLMs by a wide margin. Longer alignment training further improves results, and the two-stage strategy, i.e., first training the projection (S1), then training projection and LoRA jointly (S2), yields the strongest performance. For instance, (30K S1 + 30K S2) outperforms (100K S1), and (100K S1 + 100K S2) outperforms (500K S1) in one task and achieved comparable overall scores in our benchmarks. Switching from MAMMAL (458M parameters) to a small ESM2 (8M parameters), the performance

Table 3: Protein text generation tasks results. All scores are LLM-J

| Model | BioFM | Align. | S1 | S2 | catal. | motif | func. | prot. | Avg. |
|---|---|---|---|---|---|---|---|---|---|
| | | | 500k | 500k | **0.37** | **0.21** | **0.40** | 0.35 | **0.33** |
| BIOVERSE | MAMMAL | AR | 100k | 100k | 0.35 | 0.19 | 0.38 | 0.32 | 0.31 |
| | | | 30k | 30k | 0.32 | 0.18 | 0.33 | 0.29 | 0.28 |
| | | | 500k | – | 0.34 | 0.20 | 0.38 | 0.38 | 0.32 |
| BIOVERSE | MAMMAL | AR | 100k | – | 0.26 | 0.17 | 0.33 | 0.32 | 0.27 |
| | | | 30k | – | 0.21 | 0.11 | 0.22 | 0.31 | 0.21 |
| BIOVERSE | MAMMAL | CT | 30k | 30k | 0.33 | **0.21** | 0.39 | **0.40** | **0.33** |
| | | | 30k | – | 0.00 | 0.01 | 0.01 | 0.00 | 0.01 |
| BIOVERSE | ESM2-8M | AR | 100k | – | 0.21 | 0.12 | 0.20 | 0.24 | 0.19 |
| Granite-8B | | | | | 0.00 | 0.03 | 0.05 | 0.05 | 0.03 |
| Mixtral-8x7B | (not applicable) | | | | 0.00 | 0.02 | 0.06 | 0.02 | 0.02 |
| LLaMA-70B | | | | | 0.01 | 0.03 | 0.09 | 0.05 | 0.04 |
| GPT-OSS-120B | | | | | 0.03 | 0.09 | 0.06 | 0.10 | 0.07 |

dropped, highlighting the impact of the encoder's quality. When CT is used to reduce the training time in S1, it is important to follow it with S2, as S1-only does not teach the LLM backbone how to use those tokens in a generative task, and when combined with a prompt results in unexpected generation. However, a small S2 quickly bring up the performance of CT and with 30K S2 the performance is comparable to the longest run with AR. All results are reported in a zero-shot transfer setting. BIOVERSE is aligned (both S1 and S2) only on UniProtKB protein-text pairs with short GO terms and curated annotations, while evaluation is performed on the Mol-Instructions test split, which requires long-form, free-text property descriptions. This ensures that performance reflects transfer beyond the ontology terms used during alignment.

# 6 DISCUSSION AND FUTURE WORK

BIOVERSE demonstrates that BioFMs and LLMs can be aligned through lightweight projection layers, enabling generative reasoning across scRNA-seq, protein, and molecular modalities. This modular design allows compact LLMs to outperform much larger text-only baselines while producing richer, more interpretable outputs than candidate-matching approaches. By treating biological embeddings as first-class tokens, BIOVERSE bridges raw data and language-based reasoning in a way that is both scalable and deployable.

A key strength of BIOVERSE is its scalability across modalities: once aligned, scRNA-seq, proteins, and molecules can interoperate within the same LLM, supporting queries that span multiple levels of biology (e.g., "how does this variant protein affect cell type identity?" or "does this small molecule bind to this protein?"). Nonetheless, several limitations remain. The quality of alignment depends heavily on the underlying encoders, and modalities such as spatial transcriptomics or molecular 3D geometry are not yet explored. Current paired datasets rely largely on curated ontologies (e.g., GO terms, CellxGene metadata), which may bias reasoning and constrain coverage.

Looking ahead, several extensions are especially promising. First, interpretability can be enhanced by moving beyond single-token representations: gene-level, pathway-level, or topic-model embeddings (e.g., scETM (Zhao et al., 2021), cisTopic (Bravo González-Blas et al., 2019)) would yield more fine-grained rationales directly grounded in experimental data. Second, scaling to larger backbones (e.g., GPT-OSS-120B) and incorporating additional modalities such as epigenomics or spatial assays will test the limits of modularity and broaden biomedical applications. Third, standardized benchmarks are needed to evaluate not only accuracy but also interpretability, robustness, and factual grounding; multi-modal biological QA datasets remain scarce. Finally, integration into agentic workflows and privacy-preserving settings will be critical for real-world adoption. The design space is vast, and we have explored only a subset of configurations; further systematic ablations are essential. To accelerate progress, we will open-source our code and invite the community to co-develop multi-modal benchmarks and advance embedding-aware biomedical reasoning.

In summary, BIOVERSE offers a unified and extensible framework for embedding-aware biomedical reasoning, laying the groundwork for practical systems that connect raw scientific data with natural language understanding and interactive discovery.

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

# A    APPENDIX

## A.1    EVALUATION METRICS

We evaluate generative tasks using three complementary metrics.

**LLM-as-a-Judge** We use single-output, reference-based evaluation prompt (see below), where each model response is scored independently against the expected output. Each evaluation is repeated with 3 different random seeds (temperature = 0.2), and the average similarity score is reported.

**BERTScore** We compute semantic similarity using BERTScore with PubMedBERT embeddings, which is robust to paraphrasing and biomedical terminology variation.

**ROUGE-L** As a legacy baseline, we report ROUGE-L $F_1$ (with stemming), which measures longest common subsequence overlap between candidate and reference.

This combination captures factual correctness and coverage (LLM-Judge), semantic similarity (BERTScore), and surface-form overlap (ROUGE-L).

## A.2    PUBLIC OPEN-WEIGHTS MODELS

We list the public open-weight language models used in our experiments:

- **GPT-OSS-120B**: An open-weight, text-only model by OpenAI, available under the Apache 2.0 license. Designed for reasoning and agentic tasks.
  GitHub: `https://github.com/openai/gpt-oss`
  HuggingFace: `https://huggingface.co/openai/gpt-oss-120b`

- **Granite-8B**: We use `granite-3.3-8b-instruct`, available under the Apache 2.0 license.
  GitHub: `https://github.com/ibm-granite/granite-3.3-language-models`
  HuggingFace: `https://huggingface.co/ibm-granite/granite-3.3-8b-instruct`

- **LLaMA-70B**: Meta's open-weight large language model (Llama-3.3-70B-Instruct), available under a community license. Widely used for research and instruction-tuned variants.
  GitHub: `https://github.com/meta-llama/llama`
  HuggingFace: `https://huggingface.co/meta-llama/Llama-3.3-70B-Instruct`

- **Mixtral-8x7B**: A mixture-of-experts open-weight model released by Mistral AI, featuring 8 experts with 2 active per token. Available under the Apache 2.0 license.
  GitHub: `https://github.com/mistralai/mistral-inference`
  HuggingFace: `https://huggingface.co/mistralai/Mixtral-8x7B-Instruct-v0.1`

## A.3    TRAINING DETAILS

All experiments were conducted using the hyperparameters and configuration settings detailed below, extracted from our training scripts.

### A.3.1 MODEL & ARCHITECTURE

- **Base LLM**: `ibm-granite/granite-3.3-8b-instruct`

- **Adapter Method**: Low-Rank Adaptation (LoRA) was enabled for stage-2.

- **Input Embedding Model**: `mammal`

### A.3.2 LoRA CONFIGURATION

- **Rank (`r`)**: 16

- **Alpha (`lora_alpha`)**: 32

- **Dropout (`lora_dropout`)**: 0.05

- **Target Modules**: All linear layers (`all-linear`)

- **Bias**: `none`

- **Task Type**: Causal Language Modeling (`CAUSAL_LM`)

### A.3.3 TRAINING HYPERPARAMETERS

- **Optimizer**: AdamW (Hugging Face `Trainer` default)

- **Learning Rate**: `2e-5`

- **LR Scheduler**: Linear warmup for 100 steps, then cosine decay.

- **Weight Decay**: 0.01

- **Batch Size**: 4 per device

- **Gradient Accumulation Steps**: 1

- **Max Gradient Norm**: 1.0

- **Precision**: Mixed-precision using `bf16`

- **Seed**: 1234

- **Maximum Training Steps**: 10,000

- **Evaluation Frequency**: Every 100 steps

Experiments were run on a node with 2 NVIDIA A100 GPUs using PyTorch 2.2 and the Hugging Face Transformers library.

## A.4 PROMPTS

### A.4.1 LLM-AS-A-JUDGE PROMPT

```
You are an expert evaluator tasked with assessing the quality of a model's response to
a given instruction and input. Your goal is to compare the model's response to the
expected output and provide a similarity score between 0 and 1, where:
- 0 means the response is completely unrelated or incorrect.
- 1 means the response is perfectly aligned with the expected output.

Consider the following aspects when evaluating:
1. **Instruction Adherence**: Does the response correctly follow the instruction?
2. **Input Relevance**: Does the response appropriately use the provided input?
3. **Semantic Similarity**: Does the response convey the same meaning as the expected
output?
4. **Accuracy**: Are the facts or details in the response correct and consistent with
the expected output?
5. **Completeness**: Does the response include all key information from the expected
output?

Score Guidelines:
Use the following scale to assign a similarity score:

- 0.00      Completely unrelated, off-topic, or incorrect.
- 0.25      Minimal relevance or correctness; major elements are missing or wrong.
- 0.50      Some relevant aspects present but key points are missing or inaccurate.
- 0.75      Highly similar; closely matches the expected output with minor omissions or
differences.
- 1.00      Perfect match in meaning, accuracy, and completeness.

Instruction:
{instruction}

Input:
{input}

Model's Response:
{response}

Expected Output:
{expected_output}

Output Format (JSON):
{
  "similarity_score": <float>,
  "explanation": <string>
}
```

### A.4.2 CELL TYPE ANNOTATION INSTRUCTION-CONSTRAINED PROMPT

```
You are an expert cell biologist. Given the following information, predict the most
likely immune cell subtype. You MUST only choose from the following list of possible
cell types:
"B cells",
"CD14+ Monocytes",
"CD4 T cells",
"CD8 T cells",
"Dendritic Cells",
"FCGR3A+ Monocytes",
"Megakaryocytes",
"NK cells",
"Other"

Sorted expressed genes (most abundant first, top 128): {genes_str}
Cell embedding: [BIO]

Please respond with exactly one cell type from the list above,
with reasoning of your choice.
```

## A.5 CODE AND DATA

Code will be released upon acceptance. For anonymized review, we provide simplified code in Jupytor notebook format and a description of datasets.

### A.6 USE OF LARGE LANGUAGE MODELS (LLMs)

We acknowledge the use of large language models (LLMs) to assist in the preparation of this manuscript. The LLM was utilized for three primary purposes: (1) aiding in literature discovery to identify recent and relevant publications, (2) assisting with the formatting of LaTeX syntax for tables and equations, and (3) rephrasing sentences to improve clarity and readability. All information, including citations and scientific claims, was verified by the authors, who take full responsibility for the final content of this paper.

