# OpenReview forum: "BioVERSE: Representation Alignment of Biomedical Modalities to LLMs for Multi-Modal Reasoning"
_ICLR.cc/2026/Conference — Submitted to ICLR 2026_

### Official Review · Reviewer_mDiS · 2025-10-22

**Soundness:** 2
**Presentation:** 3
**Contribution:** 3
**Rating:** 4
**Confidence:** 3

**Summary:**

In this work, the authors train an encoder-type model for biomedical inputs both contrastively with text labels (CLIP-style) then as part of a LLM autoregressive training (VLM-style, using an adapter to soft token embeddings). In a second stage, the contrastive approach is dropped and the LLM is trained to use the input tokens generated by the biomedical encoder in more varied types of prompts.

Two use cases are considered, one for cell type classifcation based on the activation state of a set of genes for that cell, and another where the physical properties of proteins are assessed based on their chemical formulation. In both cases, some baselines are considered, and a few variations of the model training strategies are attempted, providing insights into the important elements of the training methodology.

Empirically, BIOVERSE shows strong gains on Mol-Instructions generative protein tasks, and it improves a small LLM on zero-shot, generative cell-type annotation in PBMC10K, though it trails candidate-matching approaches (e.g., LangCell) and a much larger text-only baseline.

**Strengths:**

* Clear, modular adaptation of the vision-language “encoder–projector–LLM” design to biomedical modalities. The focus on aligning BioFM embeddings directly into the LLM token space (rather than a separate joint encoder) is well motivated and practically attractive.
* Studying two alignment objectives (CT vs. AR) and a two-stage schedule adds methodological depth. The finding that CT in Stage 1 needs a short Stage 2 to be useful for generation is informative.
* Results on Mol-Instructions show sizable improvements over open-weight LLMs that see only tokenized SMILES/AA sequences, supporting the main thesis that aligned BioFM embeddings help compact LLMs generate more faithful technical text.
* The paper is globally clear about training choices (projection architecture, LoRA, token injection via inputs_embeds, ...) and openly discusses limitations.

**Weaknesses:**

* scRNA: The complex setup clearly isn't warranted in the scRNA case, since the outcome appears to be a simple classification per an ontology. The results are also disappointing, trailing a training-free GPT baseline that doesn't even receive all information (I understand that the model is bigger, but still, it doesn't justify everything).
* scRNA: A key missing baseline is a same-capacity encoder-only classifier: train the scRNA encoder (or the BIOVERSE bio encoder + MLP) directly for ontology classification. This would show whether Stage 1+2 alignment buys anything over a straightforward classifier for this use case. My guess is that it does not.
*  The article make claims about potential use cases in new label creation and explanation generation, but these claims are not substantiated at all. Another, more complex downstream task would need to be presented for these claims to stand on their own.
* I wish LLM-as-a-judge had been used for the protein/molecule properties experiment exclusively for the accuracy aspects, because right now we don't really know whether the (small) gains are caused by the addition of new information, or the learning of a soft prompt that nudges the LLM into following the instructions and style better, factors considered according to the prompt in A.4. The fact that a better encoder improves the results with this methodology is a good sign that some signals gets carried down, but not a sufficient one. It would have been nice to have another baseline for that experiment, more granular metrics, or more ablation studies.
* Confidence intervals/standard deviations are missing for most tables; LLM-J is averaged over three seeds, but variance should be reported to judge effect sizes.

**Questions:**

* Have you, since the submission, considered other downstream tasks for the scRNA experiment? If so, which ones?
* Could you report what happens when the generative model is trained with one cell type left out? Can it learn that cell type by generalizing over the examples seen, or does it only perform as a classifier?
* Would it be possible to evaluate a kNN approach for the protein properties like you did for scRNA? What: I mean, look at the protein with the most similar formulation in the training set (with some heuristic) and pretend the LLM returned its property text as the answer for the test set one? This would be useful to understand if the performance gain comes from pure memorization, or if there is also a generalization happening.
* The number of [BIO] tokens and their position within the prompt are not ablated. Given VLMs are often sensitive to these choices, it would have been good to understand how much they affect performance in these use cases.

---

### Official Review · Reviewer_AuqD · 2025-11-01

**Soundness:** 2
**Presentation:** 3
**Contribution:** 4
**Rating:** 4
**Confidence:** 4

**Summary:**

This paper introduces BioVerse, a modular framework that aligns modality‑specific biomedical encoders (for single‑cell transcriptomes, proteins, and molecules) to a frozen LLM through a small projection layer, followed by instruction tuning via LoRA. The training is explicitly two‑stage: Stage 1 (S1) performs alignment (either autoregressive with teacher forcing, or contrastive with a bidirectional InfoNCE loss); Stage 2 (S2) instruction‑tunes the LLM (via LoRA) to use the projected “soft tokens” for text generation and reasoning. The authors evaluate both retrieval (candidate matching) and generation (open‑ended text) across modalities and systematically vary AR/CT × S1/S2, yielding a clear picture of when different objectives and stages help. The architecture is lightweight i.e. requiring only small projection layers and LoRA updates, and each component can be easily replaced as newer encoders or LLMs become available.

Overall Assessment: Borderline reject (willing to accept based on author responses).
This is a strong and well-motivated paper that presents a clear contribution toward scalable, modular multimodal reasoning in the biomedical domain. The framework could solve for novel cross-modality bio tasks based on reasoning directly within the textual space. It also enables competitive retrieval models in this cross-modal space. Main reason to reject this work in its current form is to fix critically missing related work and some baselines. I am willing to revise my score, once they have a response.

Reviewer LLM Usage:
I have read the paper in full and written the review myself. Large Language Models (LLMs) were used only for writing polish, clarity improvements, and to refresh memory of (public) related work or references. The analysis and conclusions are entirely my own.

**Strengths:**

1.	Clarity and presentation: The projection‑plus‑LoRA design is simple to implement, easy to maintain, and compatible with swapping encoders or LLMs; the two‑stage split (alignment, instruction tuning) is principled and mirrors best practices in vision‑language (pre‑align then instruction‑tune).
2.	Thorough two-stage training study definition. Examples of well explored details:
   - AR and CT losses are well-motivated, clearly formulated, and empirically examined.
   - Multiple text-embedding (ϕ) strategies are compared and interpreted, with clear discussion of future pooling options to further explore (e.g., LLMEmbed, LayerAvg).
3. Retrieval + Generation combination - Unlike many single‑stage generative systems and single-modality + text systems, BioVerse explicitly optimizes a contrastive space for retrieval and also supports LLM decoding for open‑text answers. Both of these stages support three bio modalities and text.
4. Systematic discussion of results: The paper systematically studies AR vs. CT and S1 vs. S2 settings. The candidate-matching baseline is especially strong and cannot be beaten by the generative models in the first task. The authors share this result and discuss the difficulty of the generation task while explaining how it unlocks new abilities. This is insightful.
5. Embedding-space plots were also a nice add to provide clear intuition on modality alignment before and after.
6. Practical relevance: Beyond improved quantitative results, BioVerse enables novel reasoning workloads, rationale building and multi-step cross-modal interactions. It holds potential for advancing biomedical agentic systems in downstream applications.

**Weaknesses:**

1. Currently, the work risks overstating architectural originality with some missing references. The paper misses several relevant efforts that employ similar projection-to-LLM strategies, including:
 - ProteinChat (2024) and  EvoLLaMA (2024) – same design but extended by Bioverse to more modalities
 - MINT (2025, although recent this is still June 2025) – similar goal but different training objectives using preference optimization
 - OneLLM (2023) – similar approach in the general multimodality domain outside of biology.

2. Baselines seem underpowered for “best available” comparisons. The end‑to‑end “LLM‑only” baselines emphasize open‑domain models; a domain LLM such as MAMMAL is used as a component, but not to do the generative task directly (like Granite). Similarly task‑specific SOTA baselines (e.g., ProteinChat on protein generation) are missing. Finally, where feasible, a single proprietary reference model (OpenAI or Claude) on a subset would calibrate performance better.

3. Limited qualitative analysis: Demonstrations of open-ended cross-modal reasoning or question answering would strengthen claims of generalization and real-world usability in the absence of quantitative benchmarks. Additionally, a subset of these capabilities evaluated by humans would bring more confidence than just LLM-as-a-judge evaluations.

4. Results sensitivity to LoRA rank, InfoNCE temperature could be a good analysis to add.

**Questions:**

Critical for acceptance:
1. Can you explicitly position BioVerse against other works that share the same projection‑to‑LLM design (ProteinChat, EvoLLaMA, OneLLM, MINT), clarifying which elements are novel (multi‑modality span; explicit dual AR/CT objectives; two‑stage LoRA tuning) and which are incremental?

2. Are you able to add domain‑specialized generative baselines (e.g., a MAMMAL‑like bio LLM) and task‑specific SOTA comparators (e.g., ProteinChat for protein generation), and, if feasible, one proprietary LLM on a representative subset to establish an absolute reference point?

Others:
3. Can you report sensitivity to projection dimensionality, LoRA rank, and InfoNCE temperature τ, and provide run‑to‑run variance or confidence intervals for key metrics?
4. Have you done any analysis of hard negative mining (for in batch negatives) and its influence in contrastive alignment?
5. Can you complement automatic metrics with a small domain‑expert evaluation set (even N≈50 per modality) to substantiate claims about real‑world utility?

---

### Official Review · Reviewer_b3fu · 2025-11-03

**Soundness:** 2
**Presentation:** 2
**Contribution:** 2
**Rating:** 4
**Confidence:** 4

**Summary:**

The paper “BIOVERSE: Representation Alignment of Biomedical Modalities to LLMs for Multi-Modal Reasoning” proposes a unified framework that integrates pretrained biomedical foundation models (BioFMs) with large language models (LLMs) to enable joint reasoning across diverse biological data types such as single-cell RNA sequencing (scRNA-seq), proteins, and molecules. The approach, termed BIOVERSE, adapts the encoder–projector–LLM architecture commonly used in vision-language models (e.g., BLIP-2, LLaVA) to the biomedical domain.

BIOVERSE operates in two stages.
In the alignment stage, modality-specific BioFM embeddings are projected into the LLM’s token-embedding space via a lightweight multilayer perceptron (MLP), trained using either autoregressive (AR) or contrastive (CT) loss to achieve cross-modal consistency.
In the instruction-tuning stage, the system fine-tunes the LLM using curated biomedical prompt–response pairs so that the model can leverage biological embeddings in generative reasoning.

The paper evaluates BIOVERSE on multiple downstream tasks, including zero-shot cell-type annotation, molecular description generation, and protein-function reasoning, comparing it to both general-purpose LLMs (Granite-8B, GPT-OSS-120B) and domain-specific baselines. Results suggest that BIOVERSE improves alignment between biological and textual representations and enables richer generative outputs than text-only models, despite its compact size.

**Strengths:**

1. Originality

While the core idea adapts existing multimodal alignment frameworks, the paper’s attempt to extend vision–language design patterns to biomedical modalities is conceptually meaningful. It recognizes a clear gap: existing biomedical foundation models (BioFMs) remain siloed from LLM reasoning. By introducing a unified projection-based alignment mechanism, the paper brings together diverse data types—scRNA-seq, protein sequences, and molecular structures—within a single generative reasoning framework. This cross-domain unification, though incremental, shows creative adaptation to a new and technically challenging field.

2. Technical and Experimental Quality

The overall architecture is well-motivated and modular, allowing BioVERSE to plug different modality encoders into one LLM backbone without retraining. The two-stage alignment process (contrastive + autoregressive) is carefully implemented, with a clear distinction between alignment and instruction-tuning phases. The experiments, though limited, cover multiple biomedical tasks (cell type annotation, molecular description, protein function reasoning), demonstrating that the approach can generalize across data types and outperform its own backbone model. The inclusion of both small (Granite-8B) and larger (GPT-OSS-120B) LLMs also provides an initial check of scalability.

3. Clarity and Organization

The paper is clearly written and well-structured, following a logical flow from motivation to results. Figures (e.g., architecture diagrams and UMAP visualizations) effectively illustrate the alignment process and outcomes. Methodological details such as projection design, loss functions, and dataset construction are mostly transparent and easy to follow. Even though some sections could go deeper, the presentation makes it accessible to both AI researchers and biomedical audiences.

**Weaknesses:**

1. Limited Novelty

The core idea—aligning biomedical encoders with LLMs through lightweight projection layers—largely replicates the design of established vision-language frameworks such as BLIP-2, LLaVA, and Flamingo. The paper mainly transfers an existing architecture to a new domain rather than introducing genuine methodological or theoretical innovation.

2. Weak Methodology and Experimental Design

The two-stage training (contrastive + autoregressive) lacks theoretical justification and systematic ablation studies. Alignment quality is shown only through qualitative UMAP plots without quantitative metrics. The datasets used (UniProtKB, LLASmol, CellWhisperer) are narrow, curated, and ontology-biased, limiting generalizability to more complex modalities like spatial or 3D biological data.

3. Inadequate Evaluation and Argumentation

Evaluation relies heavily on the subjective “LLM-as-a-judge” metric, with no statistical significance analysis or robust baselines. Comparisons are made mostly against generic LLMs rather than domain specific models (e.g., BioT5, BioMistral, MAMMAL, ChemLLM), making it unclear whether observed gains stem from the proposed alignment or simply from pretrained BioFMs. Error analysis and interpretability validation are missing.

4. Limited Practical Impact and Presentation Quality

Despite claims of scalability and privacy-preserving deployment, no real-world demonstrations are provided. Interpretability examples are anecdotal, and important methodological details (e.g., instruction tuning, alignment configuration) are underdescribed, hindering reproducibility. Ethical and safety considerations in biomedical reasoning are also absent.

**Questions:**

see weakness.

---

### Official Review · Reviewer_NCER · 2025-11-05

**Soundness:** 2
**Presentation:** 3
**Contribution:** 2
**Rating:** 4
**Confidence:** 3

**Summary:**

BIOVERSE is a two-stage framework that aligns pretrained biomedical foundation models (BioFMs) with large language models to enable cross-modal reasoning over molecular, protein, and single-cell data. It introduces lightweight projection layers and LoRA adapters that map BioFM embeddings into the LLM token space, allowing the LLM to perform reasoning directly over BioFM-derived representations. The first stage aligns BioFM and LLM embeddings through contrastive or autoregressive objectives, and the second stage applies multimodal instruction tuning to enhance generative reasoning. Experiments indicate that BIOVERSE achieves competitive performance across multiple biomedical benchmarks and demonstrates advantages over both BioFM-only and LLM-only baselines.

**Strengths:**

This paper aligns pretrained biomedical foundation models with large language models through lightweight projection layers and incorporates LoRA fine-tuning, enabling the model to perform reasoning over multimodal biomedical information.
- While drawing inspiration from vision–language model designs, its application to biomedical modalities exhibits a certain degree of innovation.
- The methodological design is technically sound, with reasonable experimental settings and consistent, robust results across multiple datasets.
- The paper is well-structured and clearly written; with reasonable domain knowledge, readers can easily follow the content. The illustrations effectively depict the two-stage training pipeline and the cross-modal reasoning process.
- The proposed framework offers a practical approach to integrating structured biological data with LLM-based reasoning, making it relevant to researchers in the AI for Science community.

**Weaknesses:**

1. The work effectively adapts the common encoder–projector–LLM (+LoRA) paradigm to biomedical modalities and builds domain-specific alignment and instruction-tuning data. However, these adaptations are largely task wrapping and data engineering efforts rather than methodological innovations. Moreover, the proposed alignment strategy appears dated compared to recent multimodal LLMs such as Qwen3-VL, Qwen2.5-VL, and InternVL3, which employ unified pre-training or advanced token-level alignment mechanisms (e.g., token grouping, deep feature stacking).
2. While the proposed two-stage training framework is intuitively appealing, the necessity of the first stage (S1) remains insufficiently validated. The paper does not include a clear S2-only ablation. Given that the projection layer is still trainable in S2 to mitigate alignment drift, it is unclear whether the benefits attributed to S1 arise from the pre-alignment step itself or simply from the additional optimization budget.
3. The paper does not include quantitative comparisons with MAMMAL, which already unifies multiple biological modalities in a generative framework, nor with textualization-then-reasoning approaches such as TxGemma, BioT5, or Galactica. This omission limits the strength of the empirical evidence for the claimed advantage of the proposed modular alignment design.
4. The evaluation primarily involves relatively simple templated or descriptive text generation tasks (e.g., PBMC10K, Mol-Instructions). These settings assess alignment quality but do not rigorously test the model’s higher-level textual reasoning or multi-step language understanding abilities.
5. The paper provide theoretical justification that the framework supports multi-token biological embeddings: they note that multiple [BIO] tokens can be injected, define the encoder output as $z_b\in\mathbb{R}^{k\times d_b}%$, and clarify in a footnote that the framework naturally extends to k > 1. However, this has not been empirically validated — all experiments use the single-token (k = 1) setting, and the effectiveness of the model under multiple embeddings remains untested.
6. The paper claims cross-biological-modality capability and argues for advantages over MAMMAL in modularity and extensibility. However, all experiments are single-modality, without demonstrating reasoning across biological modalities. This undermines the claimed advantage, especially since MAMMAL supports true cross-modality reasoning.

**Questions:**

1. Have you explored using more recent multimodal alignment methods from state-of-the-art multimodal LLMs (e.g., unified pre-training, token grouping, or deep feature stacking)?
2. Could you provide quantitative comparisons with other methodological paradigms, such as multimodal generative frameworks and textualization-then-reasoning approaches?
3. Could you evaluate the model on more challenging language-reasoning datasets (e.g., multi-step or long-context biomedical QA and synthesis) to rigorously test higher-level textual reasoning beyond templated generation?
4. Could you include experiments that evaluate the model’s ability to perform cross-biological-modality reasoning, given that MAMMAL demonstrates such capability?

---

### Meta-Review · Area_Chair_aksj · 2026-01-09

**Summary:**

While the paper targets an important and timely problem, and the high-level framing is generally solid, the submission falls short on both conceptual novelty and empirical rigor. The proposed approach mostly reuses established multimodal alignment paradigms with not much methodological innovation, and the claimed contributions are not adequately support through convincing ablations, strong baselines, or challenging evaluations. As a result, it is difficult to attribute the reported gains to the proposed framework itself rather than to additional tuning or data curation, which significantly weakens the paper’s overall suitability for acceptance.

**Reviewer Concerns:**

N/A - no rebuttal

**Reviewer Scores:**

There was no rebuttal from the authors

---

### Decision · Program_Chairs · 2026-01-26

Reject